# Lanthanide(III) Ions and 5-Methylisophthalate Ligand Based Coordination Polymers: An Insight into Their Photoluminescence Emission and Chemosensing for Nitroaromatic Molecules

**DOI:** 10.3390/nano12223977

**Published:** 2022-11-11

**Authors:** Oier Pajuelo-Corral, Laura Razquin-Bobillo, Sara Rojas, Jose Angel García, Duane Choquesillo-Lazarte, Alfonso Salinas-Castillo, Ricardo Hernández, Antonio Rodríguez-Diéguez, Javier Cepeda

**Affiliations:** 1Departamento de Química Aplicada, Facultad de Química, Universidad del País Vasco (UPV/EHU), 20018 Donostia, Spain; 2Departamento de Química Inorgánica, UEQ, C/Severo Ochoa s/n, University of Granada, 18071 Granada, Spain; 3Departamento de Física, Facultad de Ciencia y Tecnología, Universidad del País Vasco/Euskal Herriko Unibertsitatea (UPV/EHU), 48940 Leioa, Spain; 4Laboratorio de Estudios Cristalográficos, IACT, CSIC-Universidad de Granada, Avda. de las Palmeras 4, 18100 Armilla, Spain; 5Departamento de Química Analítica, C/Severo Ochoa s/n, University of Granada, 18071 Granada, Spain

**Keywords:** coordination polymers, lanthanides, 5-methylisophthalate, photoluminescence properties, TDDFT calculations, CIS INDO/S calculations, charge transfers calculation, luminescent sensing, nitroaromatics

## Abstract

The work presented herein reports on the synthesis, structural and physico-chemical characterization, luminescence properties and luminescent sensing activity of a family of isostructural coordination polymers (CPs) with the general formula [Ln_2_(μ_4_-5Meip)_3_(DMF)]_n_ (where Ln(III) = Sm (**1_Sm_**), Eu (**2_Eu_**), Gd (**3_Gd_**), Tb (**4_Tb_**) and Yb (**5_Yb_**) and 5Meip = 5-methylisophthalate, DMF = N,N-dimethylmethanamide). Crystal structures consist of 3D frameworks tailored by the linkage between infinite lanthanide(III)-carboxylate rods by means of the tetradentate 5Meip ligands. Photoluminescence measurements in solid state at variable temperatures reveal the best-in-class properties based on the capacity of the 5Meip ligand to provide efficient energy transfers to the lanthanide(III) ions, which brings intense emissions in both the visible and near-infrared (NIR) regions. On the one hand, compound **5_Yb_** displays characteristic lanthanide-centered bands in the NIR with sizeable intensity even at room temperature. Among the compounds emitting in the visible region, **4_Tb_** presents a high QY of 63%, which may be explained according to computational calculations. At last, taking advantage of the good performance as well as high chemical and optical stability of **4_Tb_** in water and methanol, its sensing capacity to detect 2,4,6-trinitrophenol (TNP) among other nitroaromatic-like explosives has been explored, obtaining high detection capacity (with K_sv_ around 10^5^ M^−1^), low limit of detection (in the 10^−6^–10^−7^ M) and selectivity among other molecules (especially in methanol).

## 1. Introduction

In recent years, coordination polymers (CPs) and their particular subclass of metal-organic frameworks (MOFs) have awakened an increasing interest, especially for researchers developing novel multifunctional materials [1,2,3,4,5] due to their structural and chemical versatility. These features are a result of the rational design by which these materials are constructed, given the endless combinations of metal centers and organic ligands that make it possible to finally obtain the desired properties [6,7,8]. In fact, these materials have been explored in a large range of interesting applications [9], such as gas adsorption and separation [10,11,12], drug or biomolecule release [13,14], heterogeneous catalysis [15,16,17], ionic conductivity [18,19] and crystallization templates, among others [20,21]. Moreover, the presence of ordered metal ions and organic molecules in the structure imbues these materials with interesting photoluminescence (PL) properties, a fact that combined with their chemical tuneability at molecular level has also boosted their use as sensors in front of an external stimuli [22,23,24]. In this sense, it is worth specifying that the stimuli can be either a physical magnitude, such as temperature change (where luminescent thermometers provide an accurate sensitivity) [25,26,27], or a target molecule, usually promoting the attenuation of the luminescence signal [28]. Microporous luminescent CPs, provided that they present an intense signal, represent an ideal choice for fabricating devices based on luminescent sensors, given that organic ligands present in both the internal and external surface of the particles may interact with the gas molecules and modify the PL performance of the bulk material [29,30]. Regarding the PL sensing, explosive detection is, without any doubt, one of the most promising applications because of its implications in anti-terrorist operations, homeland security and environmental protection in contaminated areas [30,31,32]. Among the wide variety of explosive molecules, 2,4,6-trinitrophenol (TNP) is the most employed energetic material for the construction of improvised explosive devices, mainly due to its easy synthesis, chemical stability (because it requires a primary explosive to detonate), and accessibility from stockpiled landmines, ordnances and civilian-use explosives, all of which makes its detection a task of primary importance [33,34].

In the search and development of novel CPs with enhanced PL, those built from lanthanide(III) ions have been significantly grown during the last decade because these ions exhibit large and flexible coordination geometries and also unique luminescent properties derived from their shielded 4f electron shell [34,35]. In essence, lanthanide-based MOFs (LnMOFs) or, generally speaking, lanthanide-based coordination polymers (LnCPs), owe their luminescence to the intraionic *f-f* transitions, which are characterized by narrow and long-lived emissions in the range of near-infrared (NIR) and visible regions of the electromagnetic spectrum [36,37]. Moreover, the shielded nature of the f-electrons somehow isolates their inner transitions from the influence of the chemical environment, in such a way that they possess a characteristic emission pattern. The main advantage of LnMOFs is the improvement of the emission by the well-known antenna effect, since the lanthanide coordination to ligands may yield an efficient ligand-to-metal energy transfer that enhances the low absorption coefficients associated to these Laporte forbidden *f-f* transitions [38,39,40]. In this sense, and taking into account that the effect nurtures from the metal-ligand bond strength, ligands containing good light-harvesting groups (aromatic rings, chemical functions with lone pairs, etc.) and hard donor atoms to fulfil the HSAB principle [40] (such as carboxylate groups) are appropriate to build new materials with improved PL performance [41].

Continuing with our quest for novel materials showing enhanced PL properties, and based on the previous ideas, we report herein on five isostructural 3D CPs consisting of Ln(III) ions and 5-methylisophthalate (5Meip) ligand, named GR-MOFs15–19. These compounds present very good luminescence properties given the capacity of carboxylate linker 5Meip ligand to efficiently transfer energy to the Ln(III) centers, generating metal-centered intense emissions in both the visible and NIR with significantly high quantum yields (QY), especially for the Tb counterpart. In addition to a detailed study of their performance in solid state, involving both experimental measurements and computational calculations to unravel the PL mechanism, compounds **2_Eu_** and **4_Tb_** also prove to keep an adequate emission in liquid suspensions. Taking advantage of this fact, the sensing capacity of these materials towards a batch of solvents and nitroaromatic molecules has been studied, where **4_Tb_** presents a promising capacity in the detection of TNP.

## 2. Materials and Methods

### 2.1. Syntheses of [Ln_2_(μ_4_-5Meip)_3_(DMF)]_n_ [Where Ln(III) = Sm (**1_Sm_**), Eu (**2_Eu_**), Gd (**3_Gd_**), Tb (**4_Tb_**) and Yb (**5_Yb_**)]

All compounds were obtained by slowly dropping an H_2_O-DMF solution (10 mL, 1:1) of the corresponding lanthanide(III) nitrate hydrated salt (0.4 mmol, using 0.1778 g for Sm(NO_3_)_3_·6H_2_O, 0.1784 g for Eu(NO_3_)_3_·6H_2_O, 0.1805 g for Gd(NO_3_)_3_·6H_2_O, 0.1812 g for Tb(NO_3_)_3_·6H_2_O and 0.1869 g for Yb(NO_3_)_3_·5H_2_O) over an H_2_O-DMF solution (10 mL, 1:1) of 5-methylisophthalic acid (H_2_-5Meip) (0.6 mmol, 0.1081 g). The resulting solution was placed in a glass vessel, closed with a screw cap and placed in an oven at 140 °C. Small crystals for **3_Gd_**, **4_Tb_** and **5_Yb_** and X-ray quality single crystals for **1_Sm_** and **2_Eu_** were grown after 2 days. They were filtered off, washed with water and ethanol and dried. Yields (based on metal): 68% for **1_Sm_**, 63% for **2_Eu_**, 70% for **3_Gd_**, 65% for **4_Tb_**, 63% for **5_Yb_**. Anal. Calcd for C_30_H_25_Sm_2_NO_13_ (**1_Sm_**) (%): C, 39.67; H, 2.77; N, 1.54. Found: C, 39.43; H, 2.89; N, 1.81. Anal. Calcd for C_30_H_25_Eu_2_O_13_ (**2_Eu_**) (%): C, 39.53; H, 2.76; N, 1.54. Found: C, 39.25; H, 2.91; N, 1.78. Anal. Calcd for C_30_H_25_Gd_2_NO_13_ (**3_Gd_**) (%): C, 39.08; H, 2.73; N, 1.52. Found: C, 39.18; H, 3.03; N, 1.72. Anal. Calcd for C_30_H_25_Tb_2_NO_13_ (**4_Tb_**) (%): C, 38.94; H, 2.72; N, 1.51. Found: C, 39.10; H, 2.89; N, 1.64. Anal. Calcd for C_30_H_25_NO_13_Yb_2_ (**5_Yb_**) (%): C, 37.79; H, 2.64; N, 1.47. Found: C, 37.53; H, 2.49; N, 1.55.

### 2.2. Physical Measurements

Elemental analyses (C, H, N) were performed on a Leco CHNS-932 microanalyzer. IR spectra were acquired on diluted KBr pellets in a ThermoNicolet IR 200 spectrometer in the 4000–400 cm^−1^ spectral region. Thermal analyses (TG/DTA) were performed on a TA Instruments SDT 2960 thermal analyzer in a synthetic air atmosphere (79% N_2_/21% O_2_) with a heating rate of 5 °C·min^−1^.

### 2.3. X-ray Diffraction Data Collection and Structure Determination

X-ray data collections were performed on suitable single crystals of compounds **1_Sm_** and **2_Eu_** on a Bruker VENTURE diffractometer equipped with area detector and graphite monochromated Mo K_α_ radiation (λ = 0.71073 Å) through the ω-scan method at 130(2) K. The data reduction was performed with the APEX2 [42] software, correcting the absorption of the crystal with SADABS [43]. The crystal structures were solved by direct methods using the SHELXT program [44] and refined by full-matrix least-squares on F^2^ including all reflections with OLEX2 crystallographic package [45]. All hydrogen atoms were introduced in the difference Fourier map as fixed contributions using riding models with isotropic thermal displacement parameters of 1.2-times those of their parent atoms in both the 5Meip ligand and DMF molecules. The main crystallographic details and refinement data are gathered in Table 1.

The X-ray powder diffraction (XRPD) patterns were measured on grounded single crystals or polycrystalline samples with a Philips X’PERT powder diffractometer equipped with Cu-Kα radiation (λ = 1.5418 Å). The patterns were acquired over the 5 < 2θ < 50° range with a step size of 0.026° and an acquisition time of 2.5 s per step at 25 °C. Indexation of the diffraction profiles was made using FULLPROF program (pattern matching analysis) [46] on the basis of the space group and cell parameters obtained for single crystal X-ray diffraction of **1_Sm_**.

### 2.4. Photoluminescence Measurements

Fluorescence excitation and emission spectra and lifetime measurements on solid state were recorded on an Edinburgh Instruments FLS920 spectrometer at variable temperatures using a closed-cycle helium cryostat enclosed in the spectrometer. For steady-state measurements, a Müller-Elektronik-Optik SVX1450 Xe lamp or an IK3552R-G He–Cd continuous laser (325 nm) were used as excitation sources, whereas a microsecond pulsed µF900 lamp was used for the decay curves. The emission spectra in the NIR region and the decay curves were acquired on a Hamamatsu NIR-PMT PicoQuant FluoTime 200 detector. For the variable temperature measurements in solid state, the samples were first placed under high vacuum (of ca. 10^−9^ mbar) to avoid the presence of oxygen or water in the sample holder. For the measurements performed at room temperature on the liquid suspensions (sensing experiments), the spectra were collected on quartz cuvettes (see Luminescence Sensing Experiments section) using a Varian Cary-Eclipse Fluorescence spectrofluorometer. The photomultiplier detector voltage was set at 600 V and the instrument excitation and emission slits were open 5 nm. The quantum yield was measured in the solid state by means of a Horiba Quanta-f integrating sphere using an Oriel Instruments MS257 lamp as the excitation source and an iHR550 spectrometer from Horiba to analyze the emission. Five measurements were accomplished to properly estimate the mean and standard deviation values for each compound. Photographs of irradiated single-crystals and polycrystalline samples were taken at room temperature in a micro-PL system included in an Olympus optical microscope illuminated with a Hg lamp.

### 2.5. Luminescence Detection Experiments

For the sensing experiments conducted for compounds **2_Eu_** and **4_Tb_**, their emission capacity was first measured into different solvents by dispersing 5 mg of powder samples of the compounds into 5 mL of the corresponding solvent (water, methanol (MeOH), ethanol (EtOH), 2-isopropanol (2-isoPro), dimethylformamide (DMF), acetonitrile (MeCN), tetrahydrofurane (THF), dimethylsulfoxide (DMSO), acetone (AcO), toluene (Tol) and p-xilene). In view of the low quenching capacity of water, the dispersions for the nitroaromatics sensing experiments were prepared by adding 5 mg of powder sample of **4_Tb_** in 5 mL of an aqueous solution of 0.1 mM concentration of the corresponding nitroaromatic molecule. Quite stable suspensions were achieved after sonicating the mixture for 15 min. In view of the high sensitivity exhibited towards 2,4,6-trinitrophenol (TNP), the luminescence signal was studied according to its concentration in water. To that end, a diluted solution of TNP (0.1 mM) was added (10 μL in each addition) to the initial **4_Tb_**@H_2_O mixture and once homogenized, an emission spectrum was recorded keeping the same setup configuration. The quenching curves were quantitatively analyzed by the Stern-Volmer equation (as detailed in Appendix A), using the first suspension as a reference and calculating the concentration of the analyte from the total volume of the mixture. The detection limit was calculated according to the IUPAC recommendation of 3σ/slope, where the standard deviation σ is estimated by ten repeated fluorescence measurements of the blank dispersion of **4_Tb_**@H_2_O, and the slope value is obtained using a calibration curve of the fluorescence intensity against the concentration of TNP [47].

### 2.6. Computational Details

All Sparkle calculations were carried out using MOPAC2016 and all RM1 model for europium calculations were carried out by a modified version of the same software [48]. Calculations were performed either at the crystallographic geometry, or by fully optimizing the geometry at the particular level of theory, taking care to ensure the absence of imaginary vibrational frequencies. The Judd-Ofelt intensity parameters were calculated using the Lanthanide Luminescence Software Package (LUMPAC) [49]. 

## 3. Results and Discussion

### 3.1. Structural Description of [Ln_2_(μ_4_-5Meip)_3_(DMF)]_n_ [Where Ln(III) = Sm (**1_Sm_**), Eu (**2_Eu_**), Gd (**3_Gd_**), Tb (**4_Tb_**) and Yb (**5_Yb_**)]

The synthesized CPs are isostructural, as revealed by the analysis of the X-ray diffraction data, and crystallize in the *P*2_1_*/c* space group in the form of a three-dimensional structure. Therefore, only compound **2_Eu_** will be described in detail as a representative counterpart. The asymmetric unit consists of two europium(III) metal atoms, Eu1 and Eu2, three 5Meip ligands and a coordinated DMF molecule, which precisely concords with the neutral chemical formula of the compound. The Eu1 atom is coordinated by eight oxygen atoms, seven from carboxylate groups of 5Meip ligands and one from the DMF molecule, whereas the Eu2 atom is not coordinated to any DMF molecule, so it binds to seven carboxylate oxygen atoms (Figure 1). Both centers share the occurrence of displaying a unique chelating carboxylate moiety. According to the continuous shape measures (CShMs) performed by SHAPE program [50], the coordination sphere of Eu1 resembles a biaugmented trigonal prism (BTPR) (S_JBTPR_ = 1.936), characterized for the C_2V_ symmetry (Table 2). Instead, Eu2 atom shows a more regular pentagonal bipyramidal environment (S_PBPY_ = 1.403), which presents an ideal D_5h_ symmetry.

The three independent 5Meip ligands bind to Eu(III) ions with tetradentate modes with slightly different coordination patterns. On the one hand, two of them, denoted as A and B (see Appendix A), are equivalent and show the µ_4_-κO:κ^2^O,O’:κO’’:κO’’’ coordination mode, which contains the four-membered chelate ring. Moreover, the O3B atom is linked to two Eu1 atoms acting as a bridge between them. The unique difference between A and B is that the former is coordinated to Eu1 and Eu2 atoms separately by each carboxylate side, whereas the latter exclusively coordinates to Eu1 atoms. The third ligand of the asymmetric unit (C molecule) shows the µ_4_-κO:κO’:κO’’:κO’’’ coordination mode, in which each oxygen atom binds to one Eu2 atom (see Appendix A). The shortest intermetallic distance mediated by carboxylate groups are Eu1∙∙∙Eu1 of 4.430 Å and Eu2∙∙∙Eu2 of 4.324 Å, in which the bridging-chelating modes of A and B ligands are present. As for the three-dimensional structure, the Eu1 ions bind to each other with the B ligand and the Eu2 ions with the C ligand, building metal-carboxylate rods in both cases along the crystallographic *b* axis. The junction among the chains takes place by both B and C ligands, intertwining rods of the same Eu atom, whereas A ligands, for their part, establish the links join the Eu1-based and Eu2-based rods one another (Figure 2).

As a result, the 3D framework presents no solvent accessible voids because the potential pores are blocked by the coordinated DMF molecule. From a topological point of view, this framework may be described as a five-connected network in which both the three ligands (four-connected) and Eu(III) ions (six-connected) act as nodes, which presents as (4^2^·8^4^)(4^4^·6^2^)_2_(4^8^·6^6^·8)_2_ and belongs to the **fsy** topology.

### 3.2. Luminescence Properties

Lanthanide-based emissions in crystalline materials, such as CPs, are known to be useful for developing solid-state photodevices [25,51] given their intense emissions in the visible spectra or in the NIR region [52,53]. Therefore, photoluminescence measurements were performed on polycrystalline samples of all compounds to study their properties. 

#### 3.2.1. Photoluminescence of **3_Gd_** and H_2_-5Meip-Centred Emission Analysis

To start with, we first inspected the PL of the compound **3_Gd_** because, having no other possible inner emission line except for that at 317 nm when an UV light of 248 nm is used [54], Gd(III)-based counterpart represents faithfully the electronic structure of the ligand in the present crystalline framework. Under excitation of laser light (λ_ex_ = 325 nm), the emission spectrum presents a wide band containing two shoulders at 383 and 418 nm and the maximum sited at 505 nm (with a shoulder at 465 nm, see Appendix A). Monitoring the emission maximum, the excitation spectrum reveals the presence of a wide and band peaking at 340 nm, in agreement with the molecular nature of the ligand-centered process. In this sense, it must be remarked that both the emission spectra are substantially different to that of the free H_2_-5Meip ligand sample, since the main emission band is blue-shifted whereas the first bands (with comparatively less intensity than in the ligand) are almost maintained (Appendix A). Time-dependent DFT (TDDFT) calculations performed on a suitable model of the ligand molecule (see Computational details for further information) reproduced both spectra and revealed that the main excitation and emission of H_2_-5Meip are attributed to π-π* transitions occurring between molecular orbitals extended over the carboxylate groups and aromatic rings. Taking into account that in the structure of these Ln-MOFs the MOs localized on both carboxylate groups and π-clouds of the aromatic ring are significantly affected by the coordination to the Ln and by the π-π stacking interactions of the ligands, respectively, the observed shifts in the bands are expected (see Appendix A for detailed explanation). Seeking for a possible explanation of the observed shift, additional TDDFT calculations were also computed on a monomeric complex directly taken from the RX coordinates of compound **3_Gd_**. Although the excitation spectrum reproduces the experimental one fairly well, the emission could not be correctly simulated using singlet-to-singlet transitions, since the most intense bands found in the computed emission (λ_em_ = 414 and 475 nm) seem to correspond to shoulders of the experimental spectrum (λ_em_ = 383 and 418 nm), whereas the main experimental band (λ_em_ = 505 nm) is not observed. Seeking for a possible explanation, we turned to the idea that isophthalate derivatives are known to possess phosphorescent emissions, as proven by some CPs based on alkaline-earth ions published by us [55]. In these compounds, the phosphorescence is derived from the effective shielding of the low-energy triplet excitons occurring in the framework mainly because of the low mobility of the ligand in the framework (owing to its coordination to metal ions) and also due to the heavy atom effect [56,57]. Another emission calculation based on the previously computed excitation but using the singlet-to-triplet methodology gave rise to an emission spectrum dominated by a main band peaking at 510 nm with a shoulder at 450, representative of the main experimental band (λ_em_ = 505 nm with shoulder at 465 nm), in such a way that it supports the fact that the observed main emission derives from the triplet and not from singlet states. The MOs involved in this singlet ← triplet emission present a similar shape to those of the previously mentioned singlet ← singlet transition, which may be a possible reason for the intersystem crossing to occur. The decay curves collected on the sample at the emission maximum (λ_em_ = 505 nm) confirmed the phosphorescent nature of the main process, since the lifetime was estimated to be of 494 µs according to the exponential fitting (see Appendix A). It is worth noticing that this delayed emission is quite weak as depicted from the low intensity observed in the measurement. Moreover, assuming that this emission band concerns the lowest-lying triplet state, as dictated by Kasha’s rule [58], the energy of that state may be estimated to be at about 22,000 cm^−1^ over the ground state, which is within the range found for other related ligands. The calculation of the vertical excitation (singlet-to-triplet transition) upon the ligand molecule at the triplet state geometry gives a reasonable energy (22,912 cm^−1^) that agrees with the experimental estimation (Appendix A). These data may be relevant to explain the suitability of the ligand for the sensitization of the remaining luminescent ions, given that the triplet energy level is a key parameter in the energy transfer processes governing the luminescence of CPs (vide infra).

#### 3.2.2. Photoluminescence Performance of Visible Emitters (**1_Sm_**, **2_Eu_** and **4_Tb_**)

To follow with the compounds emitting in the visible range, a sample of **1_Sm_** was excited under laser irradiation (λ_ex_ = 325 nm) at RT to collect the corresponding emission spectrum (Figure 3). This spectrum is characterized for a similar pattern of that shown for **3_Gd_**, because the main band corresponding to the ligand emission is almost unchanged while three narrow contributions at λ_em_ = 563, 600 and 646 nm are observed. The latter bands may be attributed to Sm(III)-centered emissions assigned to ^6^H*_J_* ← ^4^G_5/2_ transitions according to the bibliography (where *J* = 5/2, 7/2 and 9/2), among which the first one, practically immersed into the ligand-centered emission, is the most intense [59,60]. In fact, when excitation spectra are recorded focusing on both λ_em_ = 600 and 646 nm, a set of narrow and intense bands (attributed to intraionic 4f transitions) arising from a wide band covering the 300–450 nm range is observed (Appendix A). It is worth noticing that the intensity of these inner transitions is notoriously different for each emission line, a fact not commonly observed in other compounds. 

In any case, the opposite situation is found in the excitation of this compound with regard to the intensity of the bands, which is characteristic of low sensitization of Sm(III) by the ligand. Surprisingly, when the temperature of the sample is cooled down to 10 K, the Sm-centered emission lines are comparatively less intense than the ligand-based pattern, which is an unexpected behavior because the antennae effect is usually enhanced with the decrease in the temperature [49,61]. However, in the present case, the cryoscopic decrease does not seem to affect the ligand-to-samarium energy transfer capacity while it substantially reduces the vibrational quenching in the ligand molecule, thus bringing an increase in the latter signal which, in turn, seems to mask the Sm-centered bands [62]. The excitation of compound **2_Eu_** under UV laser light (λ_ex_ = 325 nm) gives an emission spectrum containing, apart from a negligible band covering the 400–550 nm range, the characteristic narrow bands ascribed to the Eu-centered transitions (Figure 3). In particular, five groups of signals are observed: a single narrow band at 580 nm and multiplets centered at 592, 616, 653 and 703 nm, which are assigned to ^7^F_J_ ← ^5^D_0_ transitions (where J = 0, 1 2, 3 and 4). Among these bands, the third multiplet (^7^F_2_ ← ^5^D_0_ transition), known as hypersensitive, dominates the spectrum with an integrated intensity of more than four times that of the ^7^F_1_ ← ^5^D_0_ (222,603 vs. 50,509 counts), a fact that is in agreement with the low symmetry of the Eu sites in the crystal structure. Monitoring the main emission line, the excitation spectrum exhibits a wide band centered at 375 nm in which two contributions (305 and 340 nm) may be observed, which correspond to ligand-centered excitations. Moreover, the spectrum contains stronger narrow bands attributed to the intraionic f-f transitions, among which that sited at 395 nm is the strongest one. Cooling down the sample to 10 K brings an increase in the emission intensity (see Appendix A) that, in the present case, is not only motivated by the decrease in the vibrational quenching but also by the shift occurred in the first ligand-centered band of the excitation spectrum, in such a way that the intensity is higher at λ_ex_ = 325 nm (see Appendix A). When Tb(III) ion is coordinated to 5Meip ligands in the three-dimensional network of compound **4_Tb_**, it displays a bright green emission upon irradiation with UV light. Under monochromatic laser beam (λ_ex_ = 325 nm), the emission spectrum displays four multiplets centered at 493 nm (^7^F_6_ ← ^5^D_4_), 547 nm (^7^F_5_ ← ^5^D_4_), 588 nm (^7^F_4_ ← ^5^D_4_) and 623 nm (^7^F_3_ ← ^5^D_4_) corresponding to the mentioned intraionic transitions. With much less intensity, a fifth multiplet around 654 nm is also observed, with may be attributed to the transitions of the excited state to the lower-lying states of the ground state (^7^F_J_ ← ^5^D_4_, being J = 2, 1 and 0, see Figure 3). Among all those bands, the second one is the most intense with an intensity that quadruples that of the first band. Under a fixed emission at the most intense peak of the ^7^F_5_ ← ^5^D_4_ multiplet (λ_em_ = 542 nm), the excitation spectrum contains a wide band covering the 250–350 nm range that is assigned to the 5Meip ligand excitation, in agreement with that observed for **3_Gd_**. The absence of any Tb(III)-centered narrow lines in the excitation spectra, a priori, indicates that this ligand exerts a good antennae effect. Cooling down the sample to 10 K does not bring a significant difference in the emission but for the expected increase in the intensity derived from the decrease of the vibrational quenching (Appendix A). To further analyze the emissive properties, the decay curves were recorded at the most intense emission wavelengths to check the lifetimes of the corresponding excited states: 600 nm (^4^G_5/2_) for **1_Sm_**, 616 nm (^5^D_0_) for **2_Eu_** and 542 nm (^5^D_4_) for **4_Tb_**. All the curves show a curvilinear exponential shape (expressed in the form of log(intensity) vs. time plot) that suggests that the emission occurs from more than one radiative component, in agreement with the fact that the crystal structure contains two independent lanthanide(III) centers. Accordingly, the curves were fitted to a multi-exponential expression (I_t_ = A_0_ + A_1_exp(t/*τ*_1_) + A_2_exp(t/*τ*_2_) + A_3_exp(t/*τ*_3_)). Note that a third component had to be included in the fitting of compound **2_Eu_** to correctly reproduce the data close to the pulse of the lamp. The lifetimes were estimated by means of the weighted sum of the components, obtaining the values of 12.6(2) µs (for **1_Sm_**), 289(11) µs (for **2_Eu_**) and 806(8) µs (for **4_Tb_**). These results are in line with other previously reported CPs based on eight-coordinated Ln environments [53,61,63]. The cooling of the samples to 13 K does not affect the emission lifetimes (they are kept as 12.9(2) µs for **1_Sm_** and 816(8) µs for **4_Tb_**) except for the Eu(III) counterpart, in which the lifetime increases to 543(14) µs at low temperature. On its part, the decay curve measured for **1_Sm_** at 510 nm in the ligand emission pattern shows a lifetime of 772 ns, which remains far below that measured for **3_Gd_** (494 µs) and falls in the range of fluorescence. The reason for the shortening of the ligand-centered emission in **1_Sm_** could be due to the coupling between the triplet state and the excited states of the lanthanide(III) ion, in such a way that the charge is donated from the former to the latter partially preventing the delayed emission from the triplet to the ground state.

Additionally, the absolute emission quantum yields (QY) were also measured in the solid state at room temperature by means of an integrating sphere, using the same excitation and emission conditions as for the estimation of lifetimes. Among them, compound **4_Tb_** showed the highest QY with a value of 63(2)%, followed by **2_Eu_** with 12(2)% and a low value of 1.5(1) % for **1_Sm_**. Taking into account that the triplet state is estimated to lye at ca. 22,900 cm^−1^ over the ground state and that the emitting states of lanthanide(III) atoms are known to lye at the following energies: ^5^D_4_ for Tb(III) ≈ 20,500 cm^−1^, ^5^D_0_ for Eu(III) ≈ 17,500 cm^−1^, and ^4^G_5/2_ for Sm(III) ≈ 17,700 cm^−1^) [64], it may be stated that the system obeys Latva’s empirical rule [65]. This rule estimates that the optimal ligand-to-metal energy transfer process occurs when the afore mentioned energy gap falls in the 2500–4000 cm^−1^ range, meaning that although a direct transfer may be occurring to Tb(III), it should involve other high-lying excited states for Eu(III)- and Sm(III)-based compounds.

#### 3.2.3. Theoretical Analysis of the Luminescence on Compound **2_Eu_**

In order to better understand the luminescence properties of these isostructural compounds, **2_Eu_** has been taken as a representative example in which describe in depth the most relevant parameters governing the transfers occurring in the lanthanide/5Meip system. This analysis has been done on the basis of the experimentally recorded spectra by means of LUMPAC program [66], which has been previously used to discuss the luminescence mechanism [67,68,69,70]. To that end, we employed two models of the compound (models 2-Eu-1 and 2-Eu-2 hereafter) using the spherical atomic coordinates of the coordination excerpt of the two crystallographically independent centers (see Computational Details section). These coordinates were optimized by the Sparkle/RM1 model and charge factor (g) and polarizability (α) were estimated according to the experimental emission spectra (Appendix A). Fitting of the data by LUMPAC allowed for the estimating of the Judd-Offelt parameter as 7.93 × 10^−20^ cm^−1^ and 7.68 × 10^−20^ cm^−1^ (Ω_2_), 1.03 × 10^−20^ cm^−1^ and 1.93 × 10^−20^ cm^−1^ (Ω_4_) and 0.45 × 10^−20^ cm^−1^ and 0.06 × 10^−20^ cm^−1^ (Ω_6_), from which the intensity parameters were estimated as follows: Arad equals to 309.13 and 303.16 s^−1^ for 2-Eu-1 and 2-Eu-2, respectively, with the main contribution to the radiative decay rate by ^7^F_2_ ← ^5^D_0_ transition (78.5 and 74.6 s^−1^). Taking into account the experimental lifetime of 0.289 ms at RT, the non-radiative rate (Anrad) was determined to be of 3157.1 and 3139.3 s^−1^, respectively for 2-Eu-1 and 2-Eu-2. In view of the very similar results for both centers, we thought that the mean data could represent the system.

Another relevant parameter that describes the energy transfers occurring in this compound is the energy of the ligand’s excited states. To that end, the configuration interaction simple (CIS) of INDO/S implemented into ORCA program was employed [71]. These calculations set the singlet (S) and triplet (T) excited states around 36,000 (36,046 and 36,360 cm^−1^, respectively for 2-Eu-1 and 2-Eu-2) and 23,700 cm^−1^ (23,737 and 23,608 cm^−1^ for 2-Eu-1 and 2-Eu-2). Among these data, it is worth remarking the good fit found for the energy of the triplet state, lying at 22,912 cm^−1^. The non-radiative energy transfer rates between the ligands’ and Eu(III) excited states were also calculated by means of Malta’s models [70], which consider the occurrence of three mechanisms for the excitation of metal ions during the antenna effect: dipole-2λpole, dipole-dipole and exchange. Taking model 2-Eu-1 as representative for both centers, it could be confirmed that this compound presents an appropriate antenna effect given the large value found for the dominant triplet (T) → ^5^D_1,0_ multipolar transfers (W_ET_ being 2.44 × 10^7^ and 1.35 × 10^7^ s^−1^, respectively) when compared, for instance, with the low transfer rate observed for singlet (S) → ^5^D_4_ (W_ET_ = 9.72 101 s^−1^) (Figure 4). In addition, the back-transfer rates are significantly poor (among which the T ← ^5^D_1_ is the dominant with values of W_BET_ = 6.99 × 10^−3^ s^−1^), reaffirming the effectivity of the ligand-mediated energy transfer mechanism. Another relevant aspect to mention is that O2B(ii) atom of the Eu1 center is the greatest contributor to the radiative component of the compound, with 38% of the electric dipole mediated Arad. A close inspection of this atom shows that it pertains to the carboxylate group presenting the lowest angle with regard to the mean plane of the aromatic ring of 5Meip ligand (that is, it is most coplanar carboxylate group). This fact becomes relevant on the basis of the shape of the MO describing the triplet state of the 5Meip ligand which, as detailed above (see also Appendix A), corresponds to a π orbital extended through the whole molecule. Being that so, the carboxylate-to-aromatic ring coplanarity may bring a higher overlap of the LUMO with the Ln-based inner orbitals and hence, an improved energy transfer (see Appendix A for more details). This trend is also maintained for the ligands forming the chromophore of Eu2 center. Using all these data, the quantum efficiency is determined as 8.73% (mean value for models 2-Eu-1 (8.60%) and 2-Eu-2 (8.85%)), which serves as another indication of the goodness of these calculations given the similarity to the experimentally measured value (12%).

#### 3.2.4. Photoluminescence Performance of the NIR Emitter **5_Yb_**

At last, PL measurements were also performed for compound **5_Yb_** to check its potential emission capacity in the NIR range. The photoluminescent characterization of compound **5_Yb_** (Figure 5) revealed that, at RT and under UV laser light (λ_ex_ = 325 nm), it exhibits a similar emission profile compared to **3_Gd_**, showing a wide band with the maximum centered at 500 nm (Appendix A). This fact is, a priori, a good indication for the properties of the compound because the emission could be dominated by the population of the triplet state, a fact that could enable the antenna effect. As confirmed by the decay curve measurement at the emission maximum, it is characterized by a long-lived process with an emission lifetime of ca. 300 µs (Appendix A). Under these conditions, the bulk sample also presents a sharp emission in the NIR region that is characteristic for the Yb(III) ion, since it is majorly composed of a narrow and intense band peaking at 980 nm corresponding to the ^2^F_7/2_ ← ^2^F_5/2_ transition. This main transition is accompanied by two weak wide bands at 1005 and 1027 nm, which could be part of less probable emission of the same multiplet. The NIR emission of MOFs at room temperature is not commonly appreciated, as largely discussed earlier [72,73], because these sorts of low-energy transitions are easily quenched at room temperature. The fitting of the decay curve measured at the previous emission line gives a lifetime of 3409(8) ns. We also checked the evolution of the emission by dropping the temperature of the sample to 10 K. As observed in the Appendix A (Appendix A), the emission spectra remain almost unchanged, although there is an enhancement of the signal intensity and the two weak wide bands are better defined. Interestingly, the emission lifetime is almost duplicated, reaching a τ of 6784(9) ns (Appendix A). The measured values are somewhat short and fit within the range for other Yb-based complexes [74,75]. Therefore, these measurements raise their interest for some abovementioned specific biomedical applications in which short emissions in the NIR are needed in response to the light stimulus.

### 3.3. Sensing Experiments

The excellent PL performance revealed by compounds **2_Eu_** and **4_Tb_** in the solid state prompted us to study their sensing capacity of both toxic and explosive molecules. Taking into account that these experiments are carried out in liquid dispersions to facilitate the diffusion of the target molecules with the surface of the compounds, the solubility (using water as reference, **2_Eu_**@H_2_O hereafter) and luminescence emission capacity of these compounds were first checked. Both compounds are completely insoluble in water and maintain their structural integrity, at least by suspending the solids for 48 h, because the recovered samples keep their crystalline and unaltered PXRD diffractogram (Appendix A). Moreover, as shown in Appendix A, their emission spectra retain the intense characteristic emission bands (slightly broadened) while another wide band peaking at 393 nm (dominant for **2_Eu_**) is also observed. The appearance of the latter band, attributed to the ligand’s fluorescence, may be due to a worsening of the antenna effect in the compounds, probably caused by the interactions between ligands scaffold and the solvent. Subsequently, various suspensions were prepared using many different solvents (water, methanol (MeOH), ethanol (EtOH), 2-isopropanol (2-isoPro), N,N’-dimethylformamide (DMF), acetonitrile (MeCN), tetrahydrofuran (THF), dimethylsulfoxide (DMSO), acetone (AcO), toluene (Tol) and p-xylene), and their PL response was analyzed (Appendix A). A common feature in the emission spectra of both compounds is the fact that the intensity drop of the characteristic (lanthanide-centered) emission bands is accompanied by the increase in the ligand-centered signal, which further corroborates the abovementioned hypothesis. The two compounds show distinct solvent-dependent emission behaviors and, more interestingly, very different turning-off (evaluated as the quenching percentage of the main characteristic emission line: ^7^F_2_ ← ^5^D_0_ for **2_Eu_** and ^7^F_5_ ← ^5^D_4_ for **4_Tb_** taking H_2_O as a reference) processes derived from the solvent quenching action (Figure 6). In particular, high quenching percentages are observed for compound **2_Eu_**, that is, the emission intensity is dropped by 78-100% for all solvents, thus showing less specificity for a particular solvent. On the contrary, the turning-off process is quite diverse for **4_Tb_**, for which a high quenching effect (>75%) is only observed for Tol and AcO solvents (see Figure 6a), which also lead the quenching percentage trend for **2_Eu_**.

According to the literature, the luminescence quenching of CPs by small molecules, such as the herein studied solvents, may be caused by different phenomena, involving collapse of the framework, guest exchange, host-guest interaction, competing absorption, etc., [76]. The ligand-centered excitation band describes a distinct shape due to the different antenna effect efficiency, but shares the presence of remarkable absorption intensity in the 275–350 nm range, which is largely overlapped with the absorption band of the AcO (200–320 nm) [77]. As largely analyzed in other works [76,78], none of the remaining solvents presents such an intense absorption band above 300 nm, in such a way that the competing absorption of AcO is assumed to be the major cause for the signal decrease. However, the trend observed for the rest of solvents in the case of **2_Eu_** seems to indicate the occurrence of an additional effect. To explore the possibility of a collapse in this framework, a PXRD pattern was recorded upon a dried sample of **2_Eu_** previously suspended in AcO for 2 h. The diffractogram shows an almost amorphous phase, which confirms the low chemical stability of **2_Eu_** in most of organic solvents and explains the overall large quenching observed. Although it is not expected that two isostructural compounds present different solubility and chemical stability in solvents, this is not new since it has been already reported for other CPs [79]. In any case, despite the good sensitivity of **4_Tb_** towards AcO, no further characterization was made to study its detection capacity given the lack of enough specificity. The PL response of the compounds was also studied for the sensing of a batch of nitroaromatic molecules (1,4-dinitroluene (1,4-DNP), 3-nitrotoluene (3-NP), 4-nitrotoluene (4-NP) and 2,4,6-trinitrotoluene (TNP)), for which aqueous solutions containing them in 0.1 mM concentration were prepared. To those solutions, 5 mg of **2_Eu_**/**4_Tb_** were added and the suspension was sonicated for 15 min to get a quite stable mixture prior to the measurement. When the emission spectra were compared to the blank compound@H_2_O, we observed a similar behavior to that discussed before. In particular, **2_Eu_** revealed a significant regular quenching for the four explosives, whereas **4_Tb_** showed a more irregular pattern, presenting an almost unchanged signal for 1,4-DNP and a very high turning-off for 3-NP and TNP (with quenching percentages above 99%, see Figure 6b). In this case, it cannot be argued that the framework collapse can explain the behavior of **2_Eu_**, because it is perfectly insoluble in water, so the different quenching observed for the compounds must be related to the lower efficiency of the ligand to transfer the energy to Eu(III) centers. Given the large importance of the TNP detection worldwide, further measurements were carried out with this molecule. In essence, additional suspensions containing compound **4_Tb_** were prepared by the same method in which the concentration of TNP was gradually increased (Appendix A). As inferred from the emission spectra, all the bands experienced a rapid drop in intensity with the increase in TNP, although the hypersensitive band (^7^F_5_ ← ^5^D_4_ transition) exhibited a steeper decrease. The trend of the signal quenching in the form of a Stern-Volmer plot shows a linear relationship at low concentration, suggesting that a diffusion-controlled quenching by TNP is taking place (see insets of Appendix A). A quantitative analysis with the I_0_/I = 1 + K_sv_[TNP], where all the parameters have their usual meaning [80], gives a K_sv_ = 2.45 × 10^6^ M^−1^ in the ranges of 0–4 μM (R^2^ = 0.994). The quenching occurring in this regime may be attributed to a static quenching, which is an expected situation given the overlap between the excitation band of **4_Tb_** and the absorption of TNP [81]. Hence, it may be assumed that a fluorescence resonance energy transfer process [82] takes place by which TNP strongly absorbs light and competes with the excitation of **4_Tb_**, in such a way that the pristine effective ligand-to-terbium(III) energy transfer is weakened. To get deeper insights into this behavior, the excitation spectrum was recorded for the **4_Tb_**@H_2_O suspension containing a low TNP concentration (2.5 × 10^−6^ M). As observed in Appendix A, the excitation band is split in two and significantly blue-shifted with respect to solid state and thus more overlapped with the absorption band of TNP, which explains the large quenching generated by the latter. In contrast, at higher concentrations, the S-V plot describes a curve, a phenomenon that is usually attributed to the self-absorption by the quencher [83,84]. To properly analyze the turning-off process in the full concentration region, the curve was fitted to polynomial expression that considers both a static and a dynamic quenching (see the Appendix A for further details) [85]. The K_sv_ calculated for the whole range was estimated to be of 1.4 × 10^5^ M^−1^ (R^2^ = 0.998), which is, as far as we are aware, among the best results found for other lanthanide-based MOFs reported so far (usually in the range of 10^−4^ M^−1^, see Table 3) [30,86].

The limit of detection (LOD) is as low as 5.6 × 10^−7^ M, a fact that reveals the great capacity of this material to detect the TNP molecule. In order to further investigate the mechanism for the TNP sensing of this compound, the emission lifetime was also measured in a solution of a [TNP] = 1.0 × 10^−5^ M, estimating a τ of 367(3) µs from the exponential fitting. Such a drop in the lifetime (from 806(8) µs in solid state) is concordant with the fact that the quenching process follows not only a static but also a dynamic mechanism when the concentration of TNP is increased, probably arising from π-π interactions occurring between the TNP and ligand molecules among the external surface of the compound particles. Therefore, it can be concluded that the quenching process follows a static and dynamic mechanism. In view of the excellent sensing capacity of the compound towards TNP but the low selectivity, given the similar quenching response observed for 3-NP, we decided to explore their detection in other solvents. Among the studied ones, MeOH was selected as a good alternative not only because of its high capacity to dissolve both 3-NP and TNP, but also its relatively low quenching capacity for the emission of **4_Tb_**. In an initial comparison, the emission spectra were recorded for two **4_Tb_**@MeOH suspensions containing a small concentration of dissolved 3-NP and TNP molecules (7.5 × 10^−5^ M), finding that the characteristic terbium-based emissions are much weaker for the solution containing TNP than for 3-NP (Appendix A). Essentially, the quenching percentages for these two solutions are estimated to be of 25% and 93% for the 3-NP and TNP, which is a good starting point for a selective detection of TNP. Motivated by these results, we decided to perform a new titration experiment to analyze the quenching capacity of TNP in MeOH (Figure 7). 

A similar behavior was observed for the Stern-Volmer plot, because only the low concentration (3.32 × 10^−7^–7.98 × 10^−6^ M) region followed a linear distribution (where a K_SV_ of 3.33 × 10^6^ M^−1^ was obtained from the fitting, with R^2^ = 0.996) whereas a second order polynomial better described the whole concentration range. Best fitting results for the whole range gave a K_SV_ of 1.07 × 10^5^ M^−1^ (R^2^ = 0.991) and a LOD of 3.0 × 10^−6^ M. Although the values are comparatively lower than those found in H_2_O, the greater selectivity observed towards TNP quenching in MeOH makes the latter a more appropriate solvent for the sensing experiment. At last, in an attempt to further corroborate the sensing selectivity of **4_Tb_**@MeOH towards TNP, an additional luminescence experiment was performed. When **4_Tb_** was dispersed on a MeOH solution containing the rest of nitroaromatic molecules (a cocktail of 1,4-DNP, 3-NP and 4-NP) in a total concentration of 5.0 × 10^−5^ M, the emission signal retained most of its intensity compared to the blank MeOH solution (the intensity of the hypersensitive band remained above 75%). However, when a solution of TNP (0.1 mM) is added so that TNP reaches the same concentration in the mixture, the emission signal practically vanishes (Figure 8).

## 4. Conclusions

A family of five isostructural CPs based on Ln(III) cations and 5-methylisophthalate ligands, namely GR-MOFs15-19, has been synthesized for the first time, structurally determined and fully characterized. The crystal structure is built up through the linkage of carboxylate groups to the two crystallographically independent lanthanide(III) centers, first establishing metal-carboxylate rods which joined one another to render a quite compact building with the **fsy** topology. All compounds present good photoluminescence properties upon excitation of the ligand absorption band in the UV region. Compounds **1_Sm_**, **2_Eu_** and **4_Tb_** present emissions in the visible spectrum featuring a variable antennae effect following the **1_Sm_** < **2_Eu_** < **4_Tb_** trend, as confirmed by the quantum yields measured in solid state (1.5 < 12 < 63%, respectively). On its part, **3_Gd_** presents phosphorescent ligand-centered green emission derived from a singlet ← triplet emission, corroborated by TDDFT calculations. These calculations also allow for the estimation of the lowest-lying triplet state at ca. 22,900 cm^−1^, thus explaining the 5Meip ligand sensitization order of Tb(III) > Eu(III) > Sm(III) on the basis of Latva’s empirical rule. Moreover, CIS INDO/S calculations performed on a model of compound **2_Eu_** by means of LUMPAC software allow us to draw a key lumino-structural correlation: the higher the coplanarity of the carboxylate group with respect to the aromatic ring the higher the overlap between the LUMO describing the triplet state and the lanthanide-based inner orbitals, and hence, the higher the ligand-to-metal energy transfer. On its part, compound **5_Yb_** also presents lanthanide-centered characteristic emissions in the NIR range even at room temperature.

On the other hand, **4_Tb_** presents a promising sensing capacity towards nitroaromatic molecules, showing a preferred turning-off process when exposed to 3-NP and TNP molecules, among which the detection of TNP may be considered very sensitive in view of the high K_SV_ (1.4 × 10^5^ M^−1^) and low LOD (5.6 × 10^−7^ M) extracted from the Stern-Volmer fitting. More interestingly, when the sample is suspended in MeOH, the sensing capacity of **4_Tb_** is maintained (K_SV_ = 1.1 × 10^5^ M^−1^) but it provides a more selective detection compared to the rest of the molecules studied.

## Figures and Tables

**Figure 1 nanomaterials-12-03977-f001:**
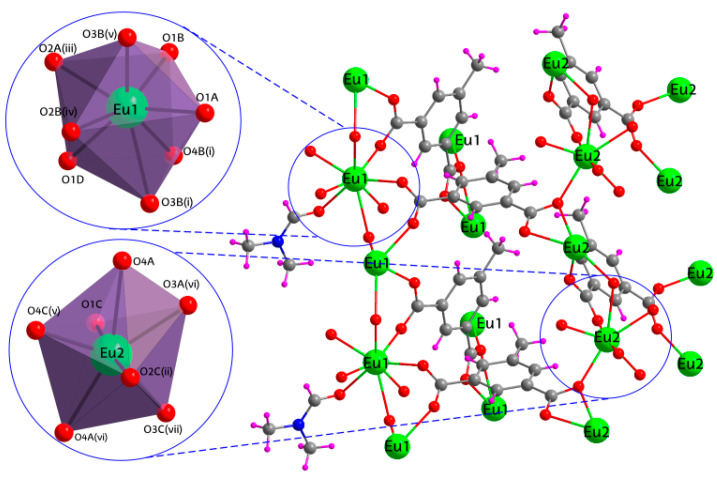
Polymeric structure and coordination polyhedra of compound **2_Eu_**. Color coding: carbon (grey), nitrogen (blue), oxygen (red), hydrogen (pink) and europium (green).

**Figure 2 nanomaterials-12-03977-f002:**
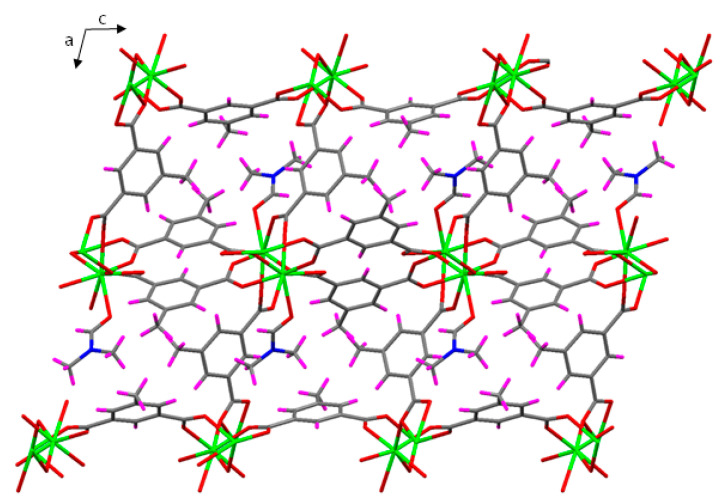
Crystal packing of compound **2_Eu_** showing the alternating propagation of the metal-carboxylate rods along *b* axis.

**Figure 3 nanomaterials-12-03977-f003:**
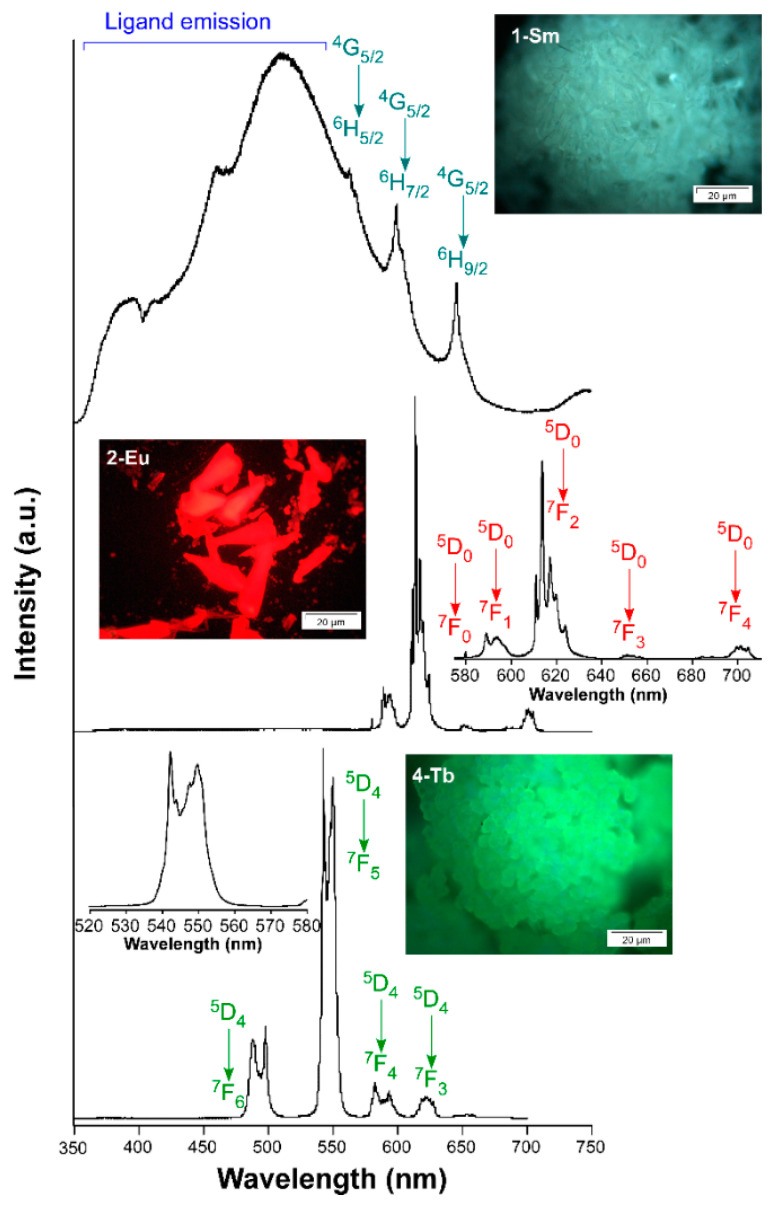
Emission spectra of compounds **1_Sm_**, **2_Eu_** and **4_Tb_** recorded at room temperature showing the intraionic assignations. Insets show selected augmented regions corresponding to the main intraionic transitions and ligand-centered emission.

**Figure 4 nanomaterials-12-03977-f004:**
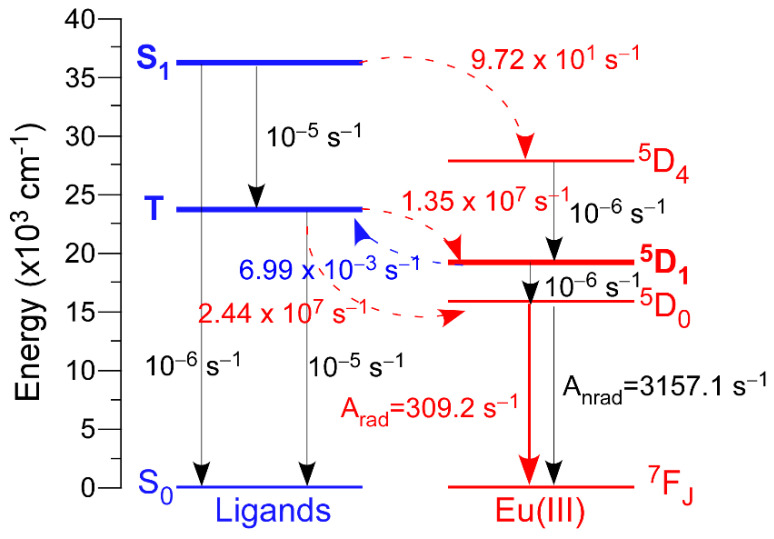
Schematic diagram of the main states involved in the luminescence of compound **2_Eu_** using model 2-Eu-1 showing the main calculated energy transfer rates. Dotted lines represent the ligand-to-metal (red) and reverse (blue) transfers.

**Figure 5 nanomaterials-12-03977-f005:**
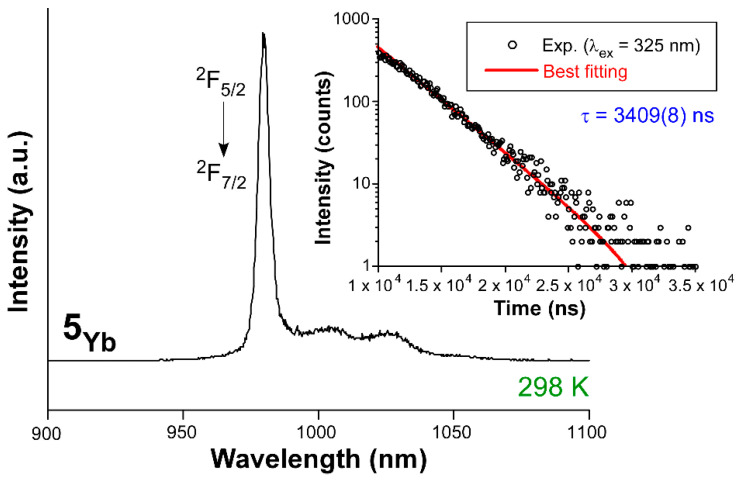
Emission spectra of compound **5_Yb_** recorded under ligand-mediated excitation in the UV region (λ_ex_ = 325 nm).

**Figure 6 nanomaterials-12-03977-f006:**
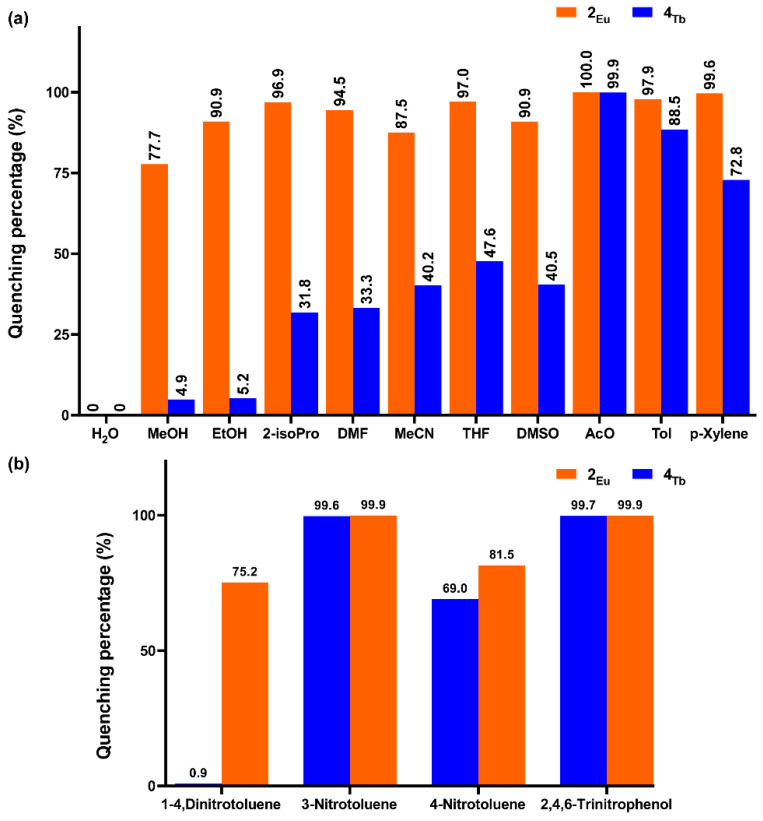
Bar charts showing the PL sensing results in the form of quenching percentages with respect to the blank compound@H_2_O dispersion for various: (**a**) solvents and (**b**) nitroaromatic molecules.

**Figure 7 nanomaterials-12-03977-f007:**
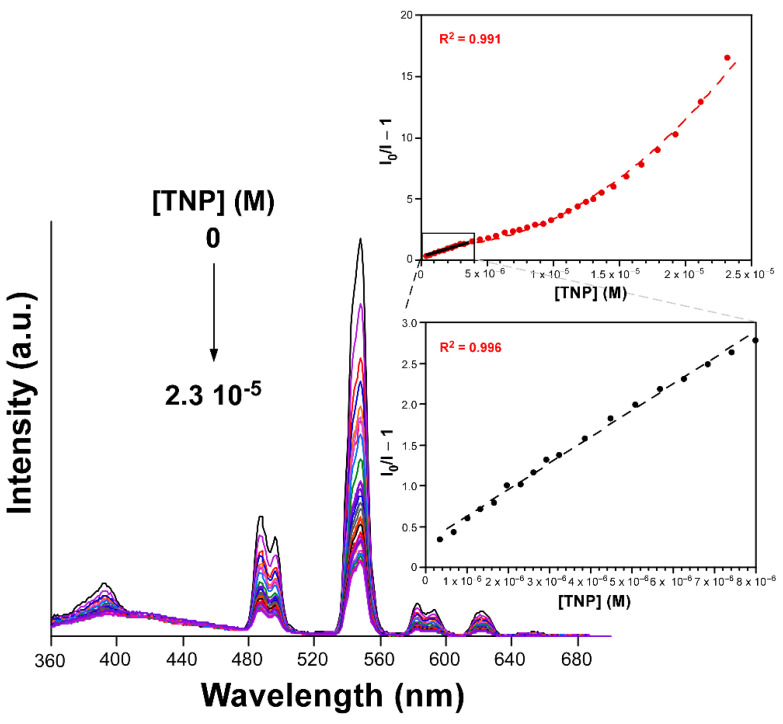
Room temperature emission spectra of **4_Tb_**@MeOH with TNP in variable concentration (λ_ex_ = 310 nm). Insets show the Stern-Volmer plots for the titration experiment with the whole studied concentration region (top) and the low concentration linear regime (bottom).

**Figure 8 nanomaterials-12-03977-f008:**
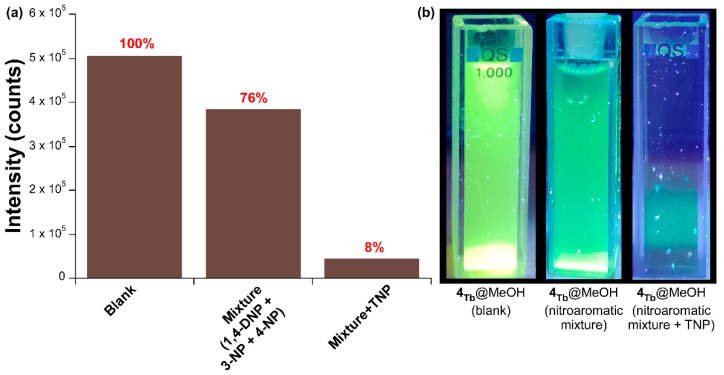
Comparison of the luminescent emission of **4_Tb_**@MeOH in the absence or presence of nitroaromatic molecules: (**a**) as a bar chart and (**b**) photographs of the irradiated suspension.

**Table 1 nanomaterials-12-03977-t001:** Single crystal X-ray diffraction data and structure refinement details of compounds **1_Sm_** and **2_Eu_**.

Compound	1_Sm_	2_Eu_
Empirical formula	C_30_H_25_Sm_2_NO_13_	C_30_H_25_Eu_2_NO_13_
Formula weight	908.21	911.43
Crystal system	monoclinic	monoclinic
Space group	*P*2_1_/*c*	*P*2_1_/*c*
a (Å)	19.662(1)	19.618(1)
b (Å)	8.3368(6)	8.3105(7)
c (Å)	18.794(1)	18.764(1)
β (°)	104.867(3)	104.959(3)
V (Å^3^)	2977.5(4)	2955.4(5)
Reflections collected	26,186	31,478
Unique data/parameters	6814/420	5215/420
Rint	0.0285	0.0424
GoF (S) ^1^	1.081	1.095
R_1_ ^2^/wR^2^ [I > 2σ(I)] ^3^	0.0521/0.1343	0.0288/0.0642
R_1_ ^2^/wR^2^ [all] ^3^	0.0608/0.1419	0.0343/0.0668

^1^ S = [∑w(F_0_^2^ − F_c_^2^)^2^/(N_obs_ − N_param_)]^1/2^. ^2^ R_1_ = ∑||F_0_| − |F_c_||/∑|F_0_|; ^3^ wR^2^ = [∑w(F_0_^2^ − F_c_^2^)^2^/∑wF_0_^2^]^1/2^; w = 1/[σ^2^(F_0_^2^) + (aP)^2^ + bP] where P = (max(F_0_^2^,0) + 2Fc^2^)/3 with a = 0.0755 and b = 15.2245 (**1_Sm_**) and a = 0.0230 and b = 11.6862 (**2_Eu_**).

**Table 2 nanomaterials-12-03977-t002:** Selected bond lengths for compound **2_Eu_** (Å) ^1^.

Coordination sphere of the Eu1 atom			
Eu1–O1A	2.353(4)	Eu1–O2B(iv)	2.335(4)
Eu1–O1B	2.300(4)	Eu1-O3B(i)	2.656(3)
Eu1–O1D	2.476(4)	Eu1-O3B(v)	2.436(3)
Eu1–O2A(iii)	2.403(4)	Eu1–O4B(i)	2.413(4)
Coordination sphere of the Eu2 atom			
Eu2–O1C	2.315(4)	Eu2–O4A	2.426(3)
Eu2–O2C	2.296(3)	Eu2–O4A(vi)	2.626(3)
Eu2–O3A(iv)	2.429(4)	Eu2–O4C(v)	2.306(4)
Eu2–O3C(vii)	2.342(4)		

^1^ Symmetries: (i) 1−x, −y, 1−z; (ii) 2−x, −½ + y, 3/2−z; (iii) 1−x, ½ + y, 3/2−z; (iv) 1−x, −½ + y, 3/2−z; (v) +x, 3/2−y, ½ + z; (vi) 2−x, ½ + y, 3/2−z; (vii) 2−x, 1−y, 1−z.

**Table 3 nanomaterials-12-03977-t003:** Results for the sensing of TNP for other lanthanide-based MOFs.

MOF	Medium	Ksv (M^−1^)	Ref.
{[Eu_2_(L_1_)_1.5_(H_2_O)_2_EtOH]·DMF}_n_	DMF	2.0 × 10^3^	[87]
{[Eu_2_(TDC)_3_(CH_3_OH)_2_]·CH_3_OH}_n_	Methanol	1.1 × 10^4^	[88]
[Pr_2_(TATMA)_2_·4DMF·4H_2_O]_n_	DMF	1.6 × 10^4^	[89]
{[Eu_3_(bpydb)_3_(HCOO)(µ_3_-OH)_2_(DMF)]·3DMF·H_2_O}_n_	Water	2.1 × 10^4^	[90]
{[Tb(L_2_)_1.5_(H_2_O)]·3H_2_O}_n_	Water	7.5 × 10^4^	[91]
[Eu(BDPO)(H_2_O)_4_]_n_	Water	6.0 × 10^5^	[90]
[Tb(BDPO)(H_2_O)_4_]_n_	Water	6.0 × 10^5^	[90]

^1^ L_1_ = 5,5′-(carbonylbis(azanediyl))diisophthalate, ^2^ TDC = thiophene-2,5-dicarboxylate, ^3^ TATMA = 4,4′,4′′-s-triazine-1,3,5-triyltri-m-aminobenzoate, ^4^ bpydb = 4,4′-(4,4′-bipyridine-2,6-diyl)dibenzoate, ^5^ L_2_ = 2-(2-Hydroxy-propionylamino)-terephthalate, ^6^ BDPO = N,N′ bis(3,5-dicarboxyphenyl)-oxalamide).

## Data Availability

Not applicable.

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
