# Peer review of "Lanthanide(III) Ions and 5-Methylisophthalate Ligand Based Coordination Polymers: An Insight into Their Photoluminescence Emission and Chemosensing for Nitroaromatic Molecules"

_nanomaterials, 2022, doi:10.3390/nano12223977_

Round 1
Reviewer 1 Report
No. nanomaterials 2002161
In this paper Antonio Rodríguez-Diéguez and coworkers reports on the synthesis, structural and physico-chemical characterization, luminescence properties and luminescent sensing activity of a family of isostructural coordination polymers (CPs), and complexes brings intense emissions in both the visible and near-infrared (NIR) regions. They also conducted DFT simulation and found the coordination environment in 2, calculations to estimate the energy of the lowest-lying triplet state and the probability of the transfers occurring in the framework allow to rationalize the emission to be responsible for the good sensing capacity . Overall, the experiment are well done, and the simulation results are useful for understanding the mechanism. In my opinion, it can be further considered for publishing in this journal. However, some revisions are needed as listed in detail below:
1. Considering the potential application and deep research, I suggest some additional experiment performed, such as mixed 4Tb-2Eu@MOF are employed to project analyses, to study it’s specificity as well, which is important.
2. The abstract section is too long, please cutoff irrelevant description, such as gas adsorption and separation drug or biomolecule release, heterogeneous catalysis, even duplicated section, and prepare a shorter and more concise one. So does the Conclusions section.
3. Generally speaking, the whole manuscript is badly organized, for example Luminescence properties of Gd, Eu and Sm compounds, tediously long discussion cannot give reader important information clearly. And the selected bond lengths for compound 2Eu, Table 3., Spherical atomic coordinates, charge factors (g) and polarizabilities …….can be placed in the supporting information. As well as page 3, ling 140, CCDC 2190996. Copies of the data can be obtained free of charge on application. I wonder if author has paid attention warring for page 5,line 208, Error! Bookmark not defined. Page 10,line 396, The observed trend for the QY may be directly connected to the energy transfer efficiency between the triplet state and the excited state of the lantha-nide(III) ion acting as charge receptor, which in turn is related to the energy gap existing between those states. This is redundant discussion.
4. in experiment section , 2.1 Synthesis should be revised into 2.1 Syntheses.
5. Products Yields should be given in details in experiment section.
6. The other suitable single crystals of compounds except Sm and Eu have not been obtained, which should be informed in text.
7. Page 4 line 163, author claimed that “All samples were first placed under high vacuum (of ca. 10–9 mbar) to avoid the presence of oxygen or water in the sample holder.” If this is, true how to apply them in under normal conditions.
8. According to Figure 2. Reader can not see clearly the Crystal packing of compound 2, and it needs redrawn.
9. Page 12, figure 5 has not referenced in this manuscript.
10. Form Figure 8 it is able to distinguish the intensity change with the concertation variation, and it should be redrawn.
11. Additional Characterization (XPS, etc.) should be given to provide the true active site and mechanism.
12. In Table S1, the numbers in the empirical formula need to be subscripted.
13. There are also a number of spelling errors, case irregularities and formatting problems in articles and graphs. Reference 78 is not identical to the others for the author list. Please examine the accuracy of words usage in whole manuscript thoroughly.
14. Some of references are out of date, such as, Pearson, R.G. Hard and Soft Acids and Bases. J. Am. Chem. Soc. 1963, 85, 3533–3539, doi:10.1021/ja00905a001, Parker, D. Excitement in f block : structure, dynamics and function of nine-coordinate chiral lanthanide complexes in aqueous media. Chem. Soc. Rev. 2004, 33, 156–165. Etc.
15 some new published relevant works should be cited to illustrate significance for the lanthanides complexes, such as…Multi-functional lanthanide-CPs based on tricarboxylphenyl terpyridyl ligand as ratiometric luminescent thermometer and highly sensitive ion sensor with turn on/off effect, Dalton Trans., 2020, 49, 4741-4750; In situ ligand-induced Ln-MOFs based on a chromophore moiety: white light emission and turn-on detection of trace antibiotic, CrystEngComm, 2022,24, 4187-4200.
Author Response
Reviewer 1:
In this paper Antonio Rodríguez-Diéguez and coworkers reports on the synthesis, structural and physico-chemical characterization, luminescence properties and luminescent sensing activity of a family of isostructural coordination polymers (CPs), and complexes brings intense emissions in both the visible and near-infrared (NIR) regions. They also conducted DFT simulation and found the coordination environment in 2, calculations to estimate the energy of the lowest-lying triplet state and the probability of the transfers occurring in the framework allow to rationalize the emission to be responsible for the good sensing capacity . Overall, the experiment are well done, and the simulation results are useful for understanding the mechanism. In my opinion, it can be further considered for publishing in this journal. However, some revisions are needed as listed in detail below:
Thank you very much for your positive evaluation of our work.
- Considering the potential application and deep research, I suggest some additional experiment performed, such as mixed 4Tb-2Eu@MOF are employed to project analyses, to study it’s specificity as well, which is important.
Response: We thank the reviewer for his/her kind suggestion and we understand that preparing this kind of heterometallic compound could be interesting for achieving a modulated luminescence (variable colour emission) response. However, we do not think the sensing response would be improved for the heterometallic Eu/Tb MOF because Eu-based luminescence turn-off does not seem to be specific to any of the tested solvents nor nitroaromatic molecules. Therefore, considering lengthy manuscript, we prefer not to add additional compounds and characterization.
- The abstract section is too long, please cutoff irrelevant description, such as gas adsorption and separation drug or biomolecule release, heterogeneous catalysis, even duplicated section, and prepare a shorter and more concise one. So does the Conclusions section.
Response: We have trimmed both the abstract and conclusions sections as required. However, in the abstract we did not discuss about gas adsorption and separation, drug or biomolecule release or heterogeneous catalysis, but there is only one sentence about the potential properties of MOFs in the introduction section. We sincerely think this general sentence describing other potential properties of MOFs is adequate to properly introduce these materials to general audience, so we prefer to keep this sentence.
- Generally speaking, the whole manuscript is badly organized, for example Luminescence properties of Gd, Eu and Sm compounds, tediously long discussion cannot give reader important information clearly. And the selected bond lengths for compound 2Eu, Table 3., Spherical atomic coordinates, charge factors (g) and polarizabilities …….can be placed in the supporting information. As well as page 3, ling 140, CCDC 2190996. Copies of the data can be obtained free of charge on application. I wonder if author has paid attention warring for page 5,line 208, Error! Bookmark not defined. Page 10,line 396, The observed trend for the QY may be directly connected to the energy transfer efficiency between the triplet state and the excited state of the lantha-nide(III) ion acting as charge receptor, which in turn is related to the energy gap existing between those states. This is redundant discussion.
Response: We do not agree with the referee in the fact that the manuscript is badly organized because we think the photoluminescence properties have been described according to a logical order, grouping the results according to the nature of the emission and making easy to draw partial conclusions in the comparison among different lanthanides emission. However, it is clear that given the large characterization conducted (and the inclusion of theoretical results), this section is very long, so we have followed his/her suggestions and moved many Table 3 and the data of the CCDC deposition information to ESI. Additionally, we have used a third-level subheading within the “3.2. Luminescence properties” section so that the reader can easily follow what is being discussed in each subsection.
On the other hand, we have removed the “Error! Bookmark not defined” sentence and we also agree with the last suggestion about the redundant discussion on the QY. Therefore, we have removed that sentence too from the manuscript.
- in experiment section , 2.1 Synthesis should be revised into 2.1 Syntheses.
Response: It has been corrected.
- Products Yields should be given in details in experiment section.
Response: We have provided the individual yields for each compound.
- The other suitable single crystals of compounds except Sm and Eu have not been obtained, which should be informed in text.
Response: We have slightly modified the sentence of the 2.1 section to better describe the obtained products obtained for each compound, as follows: “Small crystals for 3Gd, 4Tb and 5Yb and X-ray quality single crystals for 1Sm and 2Eu were grown after 2 days.”
- Page 4 line 163, author claimed that “All samples were first placed under high vacuum (of ca. 10–9 mbar) to avoid the presence of oxygen or water in the sample holder.” If this is, true how to apply them in under normal conditions.
Response: Thank you for your correction. These conditions (high vacuum) are only true for variable temperature experiments. When the compounds are measured at room temperature in normal conditions, they show an identical emission profile compared to that under high vacuum that only differs in a slight drop in the emission intensity for the former (see Figure S36).
- According to Figure 2. Reader can not see clearly the Crystal packing of compound 2, and it needs redrawn.
Response: We have modified this figure to better show the crystal packing of compound 2Eu.
- Page 12, figure 5 has not referenced in this manuscript.
Response: Thank you for your correction. We have added this reference where appropriate: “The photoluminescent characterization of compound 5Yb (Figure 5) revealed that,…”
- Form Figure 8 it is able to distinguish the intensity change with the concertation variation, and it should be redrawn.
Response: We have redrawn this figure to better show the concentration ranges used during the experiments.
- Additional Characterization (XPS, etc.) should be given to provide the true active site and mechanism.
Response: Thank you very much for your suggestion. As explained in the manuscript, we think the luminescence turn-off experimented by the compounds seems to be derived from interactions occurring between the compounds particle external surface and the nitroaromatic molecule. Unfortunately, at this moment we do not have access to such a sophisticated characterization technique because the equipment is not working. Therefore, conducting such experiments would take a long time because we should contact other research centres.
- In Table S1, the numbers in the empirical formula need to be subscripted.
Response: We understand that the referee meant to correct the point group symmetries and not the empirical formula in Table S1. The subscripts of the point symbols have been accordingly typed as subscripts.
- There are also a number of spelling errors, case irregularities and formatting problems in articles and graphs. Reference 78 is not identical to the others for the author list. Please examine the accuracy of words usage in whole manuscript thoroughly.
Response: We have correctly retyped this reference according to the style of the journal.
- Some of references are out of date, such as, Pearson, R.G. Hard and Soft Acids and Bases. J. Am. Chem. Soc. 1963, 85, 3533–3539, doi:10.1021/ja00905a001, Parker, D. Excitement in f block : structure, dynamics and function of nine-coordinate chiral lanthanide complexes in aqueous media. Chem. Soc. Rev. 2004, 33, 156–165. Etc.
Response: We have revised the references and those two mentioned have been removed. More recent references have been cited instead.
15 some new published relevant works should be cited to illustrate significance for the lanthanides complexes, such as…Multi-functional lanthanide-CPs based on tricarboxylphenyl terpyridyl ligand as ratiometric luminescent thermometer and highly sensitive ion sensor with turn on/off effect, Dalton Trans., 2020, 49, 4741-4750; In situ ligand-induced Ln-MOFs based on a chromophore moiety: white light emission and turn-on detection of trace antibiotic, CrystEngComm, 2022,24, 4187-4200.
Response: We have included the two suggested interesting references in the manuscript.
Reviewer 2 Report
This work offered by Oier Pajuelo-Corral and co-workers is very interesting because of their photoluminescence emission and chemosensing for nitroaromatic molecules. In terms of the contents, this work could be published after minor revisions as followed:
(1). Crystal data should be further refined with GoF (S) being almost equal to 1.
(2). There are too many figures and tables in the main text. Some should be transferred to SI part.
(3). The ABSTRACT and COCLUSION could be shortened.
(4). The stability in hot water experiments and organic solvents should be added. The related PXRD could be added.
(5). There are some typos and grammatical errors, the authors must make an overall revision to improve the readability. For example,
(a) “…in the NIR which are of sizeable intensity even at room temperature” should be “…in the NIR with sizeable intensity even at room temperature”
(b), “…contract with the high QY of 4Tb (of 63%)” should be “…are contract with the high QY of 4Tb (of 63%)”
(c) As for the sentence of “Time-dependent density functional theory (TDDFT) calculations and configuration interaction simple approach of intermediate neglect of differential overlap (CIS INDO/S) calculations to estimate the energy of the lowest-lying triplet state and the probability of the transfers occurring in the framework allow to rationalize the emission capacity of the compounds” should be “shows good electrochemical characteristics, which are proven by ” It is too long, few people could understand. Authors should check the text and redescribe all long sentences.
(6). In the introduction part, some references should be added, such as ACS Catal. 2022, 12(16), 10373–10383. Mater. Today Chem., 2022, 24, 100984. In addition, some old references should be removed. Moreover, more references from Nanomaterials should be cited.
Author Response
Reviewer 2:
This work offered by Oier Pajuelo-Corral and co-workers is very interesting because of their photoluminescence emission and chemosensing for nitroaromatic molecules. In terms of the contents, this work could be published after minor revisions as followed:
Thank you very much for your positive evaluation of our work.
(1). Crystal data should be further refined with GoF (S) being almost equal to 1.
Response: We are aware that, in theory, the structure refinement of a perfect crystal should have an S value equal to 1, but in practice, such a value is not always possible for most of crystals, at least in the case of MOFs, because these compounds usually crystallize with a big number of defects. In any case, we have redone data reduction and structure refinement of crystal 2Eu because the S was above 1.1 and we thought it could be slightly improved. After all our best efforts, we have omitted some few (outliner) reflections and carried out additional refinement cycles, achieving the actual converged structure with a S = 1.095. We believe the actual refined structure is of good quality as to be published.
(2). There are too many figures and tables in the main text. Some should be transferred to SI part.
Response: We have moved some figures and tables (Figure 7, Table 3) as well as some other less important parts (i.e. CCDC deposit information) to the ESI, in such a way that the actual revised version has been substantially trimmed.
(3). The ABSTRACT and COCLUSION could be shortened.
Response: Both the abstract and conclusion sections have been substantially trimmed, keeping solely the most important ideas.
(4). The stability in hot water experiments and organic solvents should be added. The related PXRD could be added.
Response: The stability of compound 4Tb was already confirmed by PXRD data, both collecting the powder from water and after the experiment of the sensing of TNP. These results are gathered in Figures S36 and S39.
(5). There are some typos and grammatical errors, the authors must make an overall revision to improve the readability. For example,
Response: Thank you very much for your corrections, we have done them. The whole manuscript has been carefully revised to correct all typos and grammatical errors.
(a) “…in the NIR which are of sizeable intensity even at room temperature” should be “…in the NIR with sizeable intensity even at room temperature”
Response: It has been corrected.
(b), “…contract with the high QY of 4Tb (of 63%)” should be “…are contract with the high QY of 4Tb (of 63%)”
Response: This sentence in the abstract has been retyped to shorten the abstract content.
(c) As for the sentence of “Time-dependent density functional theory (TDDFT) calculations and configuration interaction simple approach of intermediate neglect of differential overlap (CIS INDO/S) calculations to estimate the energy of the lowest-lying triplet state and the probability of the transfers occurring in the framework allow to rationalize the emission capacity of the compounds” should be “shows good electrochemical characteristics, which are proven by ” It is too long, few people could understand. Authors should check the text and redescribe all long sentences.
Response: This sentence has been removed from the abstract.
(6). In the introduction part, some references should be added, such as ACS Catal. 2022, 12(16), 10373–10383. Mater. Today Chem., 2022, 24, 100984. In addition, some old references should be removed. Moreover, more references from Nanomaterials should be cited.
Response: These two references have been included and some old references have been replaced by more recent references from Nanomaterials journal.